# Effects of Hydrothermal Aging of Carbon Fiber Reinforced Polycarbonate Composites on Mechanical Performance and Sand Erosion Resistance

**DOI:** 10.3390/polym12112453

**Published:** 2020-10-23

**Authors:** Mei Fang, Na Zhang, Ming Huang, Bo Lu, Khalid Lamnawar, Chuntai Liu, Changyu Shen

**Affiliations:** 1National Engineering Research Center for Advanced Polymer Processing Technology, Zhengzhou University, Zhengzhou 450002, China; 16603869720@163.com (M.F.); huangming@zzu.edu.cn (M.H.); bolu@zzu.edu.cn (B.L.); ctliu@zzu.edu.cn (C.L.); shency@zzu.edu.cn (C.S.); 2Ingénierie des Matériaux Polymères, UMR 5223, CNRS, INSA Lyon, Université de Lyon, Villeurbanne 69622, France; khalid.lamnawar@insa-lyon.fr; 3State Key Laboratory of Structural Analysis for Industrial Equipment, Dalian University of Technology, Dalian 116023, China

**Keywords:** carbon fiber reinforced polycarbonate composites, hydrothermal aging, solid particle erosion, mechanical property

## Abstract

Carbon fiber reinforced polycarbonate (CF/PC) composites have attracted attention for their excellent performances. However, their performances are greatly affected by environmental factors. In this work, the composites were exposed to hydrothermal aging to investigate the effects of a hot and humid environment. The mechanical properties of CF/PC composites with different aging times (0, 7, 14, 21, 28, 35, and 42 days) were analyzed. It was demonstrated that the storage modulus of CF/PC composites with hot water aged for seven days has the highest value in this sampling period and frequency. Through the solid particle erosion experiment, it was found that the hydrothermal aging causes the deviation of the maximum erosion angle of composites, indicating the composites underwent ductile–brittle transformation. Furthermore, the crack and cavity resulting from the absorption of water was observed via the scanning electron microscope (SEM). This suggested that the hydrothermal aging leads to the plasticization and degradation of CF/PC composites, resulting in a reduction of corrosion resistance.

## 1. Introduction

Carbon fiber reinforced polymer composites (CFRP) have been widely used in construction, transportation, and sports equipment fields, profiting from their high durability and strength, light weight, and excellent thermodynamic stability [1,2,3,4,5]. Zhang et al. [6] demonstrated that the average tensile and compression strength of glass fiber (GF) composites is almost 50% lower than CF composites. Batuwitage et al. [7] reported that the strength of CFRP composite is ten times as much as steel at the same volume.

Nevertheless, the properties of CFRP composites will change during their operating life because of the complex work environment [8,9,10]. Many works have been published about the aging resistance of CFRP. Some researches demonstrated that the matrix and interface of composites are greatly influenced by hydrothermal effect [11,12]. Generally, there are three types of composites absorbing water: transporting of water molecules within the matrix, permeating at the matrix–fiber interface, and absorbing into the crack produced by the action of high temperature [13,14,15]. The entry of water molecules causes the matrix to expand, which induces the residual or hydrothermal stress in CFRP composites [16]. Moreover, moisture accelerates the degradation of composites through the breaking up of molecular chains and deterioration of the interface of matrix/fiber [17,18]. High temperature environments accelerate the diffusion rate of moisture and result in the plasticization of CF/PC composites [8,11,19].

In many industrial applications, sand erosion is of wide concern due to its serious friction drag, structural integrity, and high maintenance costs, e.g., in helicopter rotor blades and high-speed vehicles, whose surfaces are usually exposed to dusty environments (contain erodent flux conditions, erosive particle characteristics). Extreme conditions, such as sand and dust environments, may even accelerate erosion and wear processes [20]. Hydrothermal aging also has an important effect on impact property for composites. Lu et al. [21] investigated the impact behavior of unidirectional CF- reinforced epoxy resin compounds after hydrothermal aging and found that the entry of water molecules decreased the impact resistance of CF/PC compounds. Ahmad et al. [22] concluded that the water molecules have a negative effect on the impact resistance of composite plate. Hanan et al. [23] found the impact damage for CF/PC composite was influenced by hydrothermal aging via ultrasonic technology. Based on the above-mentioned factors, it is extremely urgent to explore the effects of hydrothermal aging on the mechanical properties and solid particle erosion resistance for CFRP composites. 

In this study, polycarbonate (PC) was chosen as matrix for its excellent fatigue resistance. The aging process of CF/PC composites was carried out under a humid and hot environment. The moisture absorption rate was measured and calculated during the aging process. The changes of mechanical properties before and after hydrothermal aging were measured through tensile and flexural tests. Underlying mechanisms for hydrothermal aging induced changes in mechanical properties and sand erosion property were surveyed. Meanwhile, the effect of hydrothermal aging on thermodynamic stability of CF/PC was examined by dynamic mechanical analysis (DMA). The surface morphologies of CF/PC were analyzed through SEM and three-dimensional hyper depth of field. It is expected that, in the case of CF/PC material being chosen as a surface material, hydrothermal aging protection will be applied, so as not to affect the product performance.

## 2. Experimental

### 2.1. CF/PC Composites

T700SC-3K Carbon fiber (CF) supplied by Covestro Company (Tokyo, Japan) is adopted as reinforcing CF/PC composites. The Makrolon series of Polycarbonate (PC) 2407 was purchased from Covestro Company in Germany. Figure 1 shows the pictures of the lamination diagram and its preparation process. The specific preparation process is as follows: Liquid PC infiltrated CF and solidified to make CF/PC unidirectional single layer belt. According to the design requirements, the single layer belt was cut at different angles in the light of the laying direction of CF. To avoid the warping deformation of products, 8-layer unidirectional tapes were assembled at angles of 0°/90°/+45°/−45°/−45°/+45°/90°/0°. In the end, CF/PC composites were prepared and molded by hot processing at 240 °C for 3 min.

### 2.2. Hydrothermal Aging

Samples were divided into seven groups and immersed in deionized water at 80 °C. The aging time of the seven groups was 0, 7, 14, 21, 28, 35, and 42 days, respectively. After hydrothermal aging, specimens were taken out and dried for 24 h at 25 °C.

### 2.3. Moisture Absorption Test

Seven specimens of un-aged were selected and marked, each specimen was weighed by 0.1 mg accuracy electronic scale and labeled as *M_0_*_._ The sample weight was measured regularly during the aging process at the following time interval: every day in the earlier stage and every seven days in the later period. During testing, specimens were taken out, dried, weighed, and put back. Moisture uptake was measured from the average weight at time *t*, and moisture absorption, *W_t_*_,_ is defined in the following equation.
(1)$$W_{t}\left(\%\right)=\frac{M_{t}-M_{0}}{M_{0}}\times 100$$where *M_0_* is the initial weight and *M_t_* is the weight at time *t*.

### 2.4. Mechanical Performance Testing

Universal tensile tester (model INSTRON 5585, Boston, MA, USA) was applied to study the tensile property of CF/PC of 170 mm × 12 mm × 2 mm dimension according to GB-T 1040, the tests were performed at room temperature (RT). The bending tests were conducted using an INSTRON 5585 machine at a span length of 31.5 mm according to GB-T 9341-2000. The tensile and bending tests were performed at the same rate of 1 mm/min. At least seven specimens were measured for each test and the average value was derived.

### 2.5. Dynamic Mechanical Analysis (DMA)

The DMA analyzer (Q800) was used to characterize the dynamic thermo-mechanical properties of un-aged and hydrothermal aged CF/PC specimens. The single cantilever mode was used during the test. The specimens with size of 30 mm × 12 mm × 2 mm were heated from 50 °C to 200 °C at a heating rate of 3 °C/min with a frequency of 1 Hz. At least five specimens were scanned for each sample.

### 2.6. Solid Particle Erosion 

STR-9060 model sand-blasting equipment (Zhangjiagang Stell Coating Equipment Co.LTD, Suzhou, China) was employed to explore the influence of hydrothermal aging on the specimen erosion resistance. The erodent was silicon carbide (SiC) with sharp edge, and the average size was 300 µm to 800 µm. The SiC particles impinged on the sample surface under the acceleration of high pressure gas. The mass flow of the particles is 16.7 g/s in 0.345 MPa pressure. The distance between the specimen holder and the nozzle was 30 mm and the impact angle was 30°, the inner diameter of the nozzle is 6 mm, and each specimen was eroded for 1 min. After erosion, in order to clear away the SiC particles, specimens were washed by ethyl alcohol and dried by air blasting. The weight loss was measured and calculated by at least seven specimens. All tests were done at RT.

### 2.7. Scanning Electron Microscope (SEM)

The surface morphology of specimens with different aging time was analyzed by a JEOL JSM-7500F (Tokyo, Japan) scanning election microscopy (SEM). A thin layer of gold was sprayed on the surfaces of specimens to make them more conductive and visible.

## 3. Results and Discussion

### 3.1. Moisture Absorption Analysis

The relationship between water absorption and aging time was displayed in Figure 2. The carbonate base of PC molecular chain with strong polarity could interact with water molecules [24,25,26], allowing the CF/PC composites to absorb moisture. As clearly seen in Figure 2, the moisture absorption is increasing linearly with the square root of five days of aging. Of note, the specimens approached the saturation point after five days, and the water absorption capacity decreases from the fifth to seventh day, when it reaches water absorption saturation state. After seven days, the hygroscopic equilibrium line reached a flat state.

Figure 3a presents a smooth and compact surface for un-aged specimen. Surprisingly, after 42 days, the surfaces of the hydrothermally aging samples (Figure 3b) showed deep cracks and small debris. Clearly, water molecules entered CF/PC composites along the interface of the fiber matrix and occupied additional volume. Once the water molecules evaporate from the composite, cracks and voids appear both inside and on the surface.

### 3.2. Tensile Property Analysis

Representative tensile behavior of CF/PC at different aging times is displayed in Figure 4. The elongation at break (from Figure 4a) of CF/PC decreased with the aging time. On the premise of sampling period and frequency every seven days, the tensile strength (Figure 4b) of specimens reached its peak on the 7th day of aging in hot water at 80 °C. A tighter structure owing to the entanglement of molecular chains in a hot environment is beneficial to the improvement of tensile properties.

However, after seven days aging, the tensile properties of CF/PC decreased rapidly with the increasing of aging time. It was also found that the tensile strength of CF/PC after 14 days aging was inferior to virgin samples. The maximum tensile stress of the CF/PC composites decreased slightly. There are two reasons for the decline in mechanical properties of CF/PC composites: the presence of absorbed water and the degradation of CF/PC composites.

Polycarbonate could react with water molecules due to their strong polarity. The schematic diagram of hydrolysis mechanism for PC was seen as Figure 5. Hydroxyl groups in water molecules bind to PC chains, leading to the breaking of molecular chains and hydrolysis of the polymer [27]. Macroscopically, the polymer performance decreases.

### 3.3. Three-Point Bending Analysis

Figure 6 shows the flexural property of CF/PC composites as a function of aging time. It can be seen from Figure 6a–c that the flexural performance of specimens increased first and then decreased with the aging time. On the premise of sampling period and frequency every seven days, the maximum bending performance was at 14 days for hydrothermal aging. Figure 6d displayed that the flexural strain declines as the aging time increases. According to the result of three-point bending, the tensile strength and flexural strength values are very close. This is because, for the test of flexural strength, the upper part of the specimen is compressed and the lower part is stretched. The lower part subjected to stretch damaged first, indicating that the material is brittle. Combined with the result of sand erosion in Figure 8, it is shown that the ductile–brittle transition occurs in CF/PC composites.

On one hand, the matrix of CF/PC would shrink when aged at 80 °C, resulting in a closer attachment with CF and matrix. In addition, the residual stress of CF/PC composites was eliminated, and the regularity of CF/PC composites structure was improved when aged in high temperature environment. Thus, the flexural property increased in the early aging. On the other hand, the absorption of water molecule caused voids and defects within CF/PC composites. The matrix would separate from carbon fiber owing to the swelling effects. On the whole, the flexural strength and modulus decreased with further aging.

### 3.4. Dynamic Thermal Mechanical Analysis

Figure 7 presents the storage modulus variation with the aging time. Results reveal an increase of composites modulus for seven days aging. However, the moduli of CF/PC composites aging for 14 days to 42 days are lower than those un-aged. This results further confirmed the results in Figure 6. In the early aging, composites structure became tight, facilitating the increase of modulus. During the later aging time, the water molecules caused cavities and cracks in the matrix of CF/PC composites.

### 3.5. Solid Particle Erosion Performance

Solid particle erosion test was performed on CF/PC for the purpose of studying the sand erosion behavior. Under the same impact conditions, the weight loss of CF/PC composites increased and then decreased (Figure 8). Compared with un-aged composites, the samples aging for seven days exhibited the worst sand erosion resistance performance.

The un-aged CF/PC composites were severely eroded at 30°, and the weight loss decreased at an erosion angle of 60°. Nevertheless, the maximum erosion behavior of plastic material occurs at a low angle [28]. Therefore, it could be concluded that the PC matrix indicated a ductile characteristic after hydrothermal aging (7–42 days).

With the increase of hydrothermal aging time, water molecules occupied more additional volume inside the CF/PC composite, resulting in the structure become compact and shown the characteristic of brittleness. Hence, the maximum erosion angle of CF/PC composites has changed. Therefore, the weight loss of CF/PC composites aging for 14 to 42 days decrease compared with un-aged CF/PC when eroded at 30°. Conversely, the weight loss of CF/PC composites after hydrothermal aging (7–42 days) increased obviously when eroded at 60°. This result further proves the ductile and brittle transition of the CF/PC composites during hydrothermal aging. 

### 3.6. Surface Morphology Analysis

The eroded surfaces of un-aged CF/PC composites were presented in Figure 9a,b. A mass of matrix fragments was attached to the fiber, indicating a perfect combination of fibers and matrix. Large amounts of cracks and matrix fragments were on CF/PC surface, and a few fibers are exposed and broken. Under a 30° erosion angle, CF/PC composite is subjected to significant shear stress. Repeated cutting of high-speed particles will cause deformation, scratches, and pits on CF/PC surfaces, causing a weight loss of CF/PC composite. In addition, some tiny particles embedded in the cracks would accelerate the crack propagation and mass loss of CF/PC composites.

Figure 9c–f displayed that the matrix stripping fiber fracture and crack propagation occurred on aged specimens under the erosion of sand. Compared to un-aged composites, a small amount of matrix was attached to the fiber surface (see Figure 9c,d) and the fiber (see Figure 9e,f) has a smooth surface with no matrix attached. On one hand, the infiltration of water molecules increases the distance between the molecules of the CF/PC composites. On the other hand, the combination between water molecules and PC matrix shielded the interaction of CF and matrix. The eroded surface of CF/PC composites aging for seven days are shown in Figure 9c,d. When CF/PC was eroded at the angle of 30°, deep pits and broken fiber can be clearly observed. Under the action of hot water aging, the terminal segments of the PC chemical groups are easily degraded, resulting in plasticization of the matrix, which is manifested as the decrease of the sand erosion resistance of CF/PC composites at the macro level. 

Combined with Figure 8 and Figure 9e,f, it was found that CF/PC samples aged 42 days eroded at of 30° was the lowest. Exposed to hot and humid environment for a long time, the CF/PC would undergo ductile-brittle transition, and its maximum erosion angle transfers to 90°, the vertical impact of solid particles is the most serious damage to the material [30]. The particles generated less vertical force when impinging on the sample at 30°, and therefore less erosion was caused to the sample.

The three-dimensional morphology on CF/PC surface after solid particle erosion was exhibited as Figure 10. Lots of scratches formed on the surface of un-aged samples after sand erosion (see Figure 10a). The height from crest to trough was 0.46 mm. Compared with un-aged samples, extensive pit was formed on the surface of CF/PC composites aged for seven days under the solid particle erosion. And the depth of the pit reached 0.62 mm (Figure 10b). The CF/PC composites show a poor solid particle erosion resistance after 42 days hydrothermal aging. As shown in Figure 10c, there are a lot of ‘lips’ on the CF/PC composites surface. However, the weight loss of the CF/PC composites is less, indicating that the shear stress has little influence on the erosion of the CF/PC composites. It was also indicated that the CF/PC composites undergo a ductile–brittle transition and the maximum erosion angle has changed.

## 4. Conclusions

This work presents the effects of hydrothermal aging on the mechanical properties and erosion resistance for CF/PC composites. The moisture absorption rate of CF/PC composites grows linearly with the square root of aging time during the first five days, and it stays flat when reaching saturation. On the premise of a sampling period and frequency every seven days, the tensile properties reached their maximum at the seventh day and the peak value of bending flexural performance was reached on the 14th day. The effects of aging time on storage modulus and solid particle erosion resistance are consistent with the stretching results. Hydrothermal aging causes holes to form on the surface of CF/PC composites and reduces the specimen’s sand erosion resistance. Meanwhile, the physical property of CF/PC composites was changed.

## Figures and Tables

**Figure 1 polymers-12-02453-f001:**
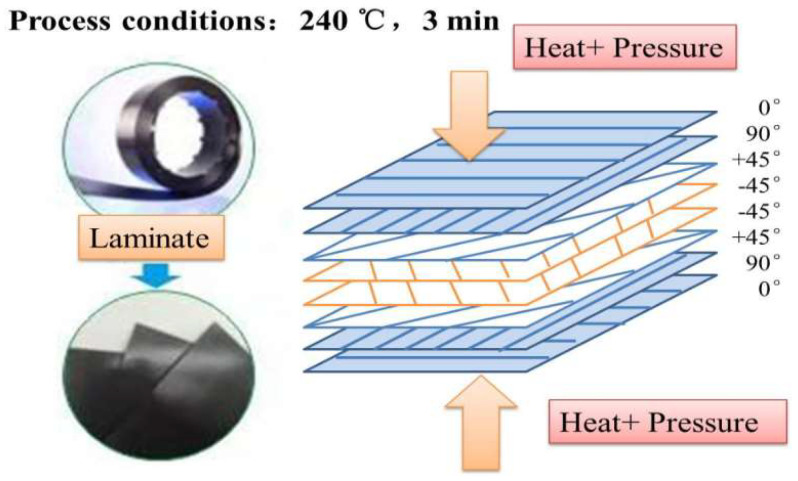
Schematic diagram of forming process for prepreg laminate.

**Figure 2 polymers-12-02453-f002:**
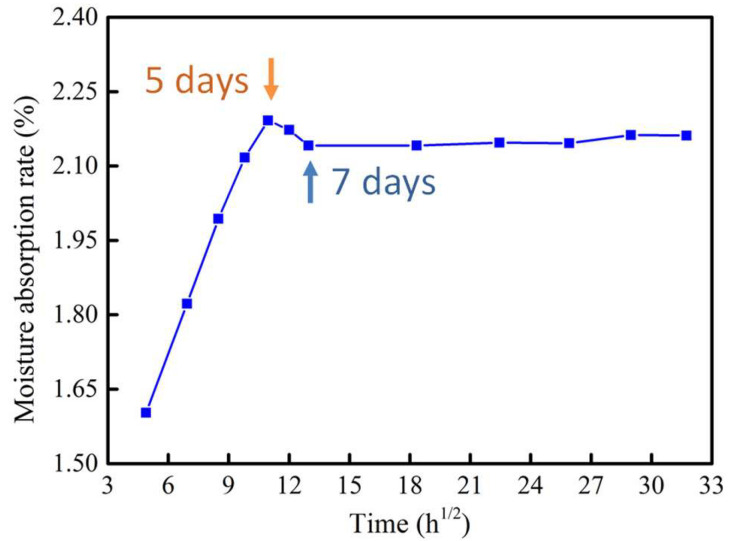
Water absorption of CF/PC composites.

**Figure 3 polymers-12-02453-f003:**
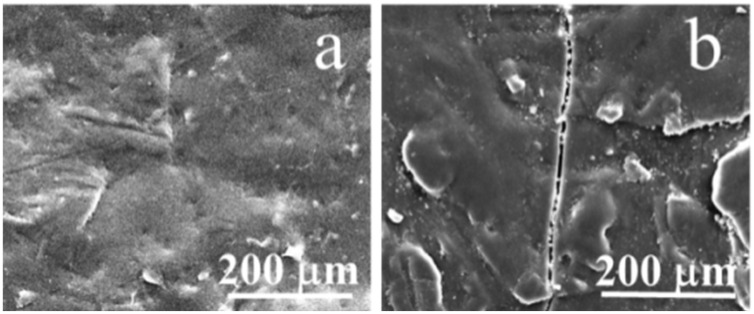
Surface topography of CF/PC composites. (**a**) un-aged composites surface, (**b**) hydrothermal aged for 42 days surface of CF/PC composites. 80 °C.

**Figure 4 polymers-12-02453-f004:**
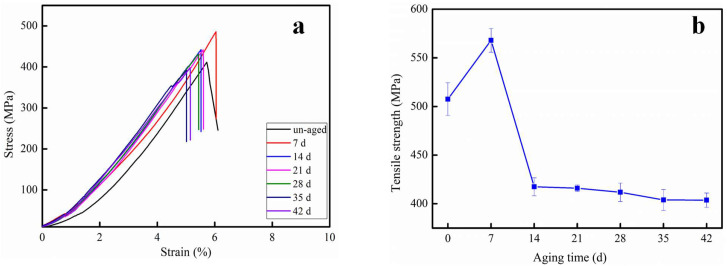
Tensile properties for CF/PC composites. (**a**) the representative stress strain curves of CF/PC composites with different hydrothermal aged, (**b**) tensile strength of CF/PC composites with different hydrothermal aged. 1 mm/min.

**Figure 5 polymers-12-02453-f005:**
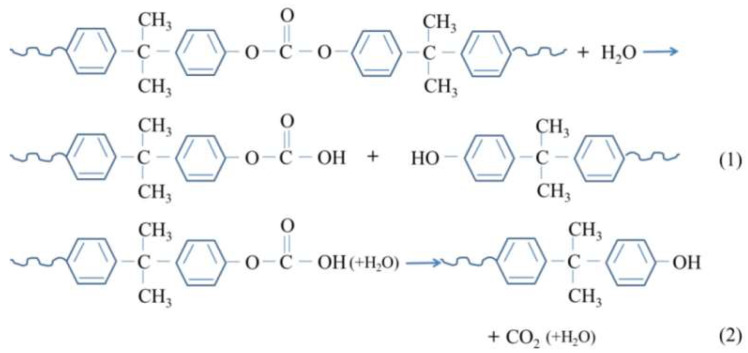
Schematic diagram of hydrolysis mechanism for PC.

**Figure 6 polymers-12-02453-f006:**
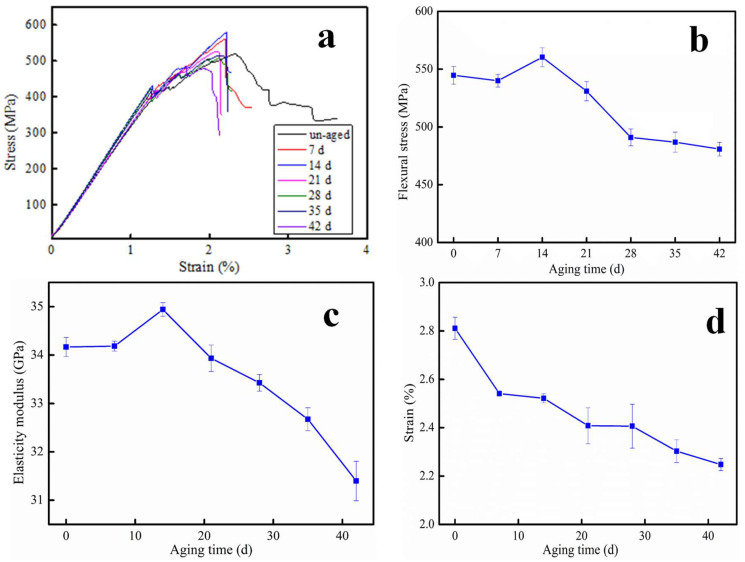
Flexural properties for CF/PC composites as a function of aging time. (**a**) the representative stress-strain curves of CF/PC composites with different hydrothermal aged, (**b**) flexural stress of CF/PC composites with different hydrothermal aged, (**c**) elasticity modulus with different hydrothermal aged, (**d**) strain with different hydrothermal aged. 1 mm/min.

**Figure 7 polymers-12-02453-f007:**
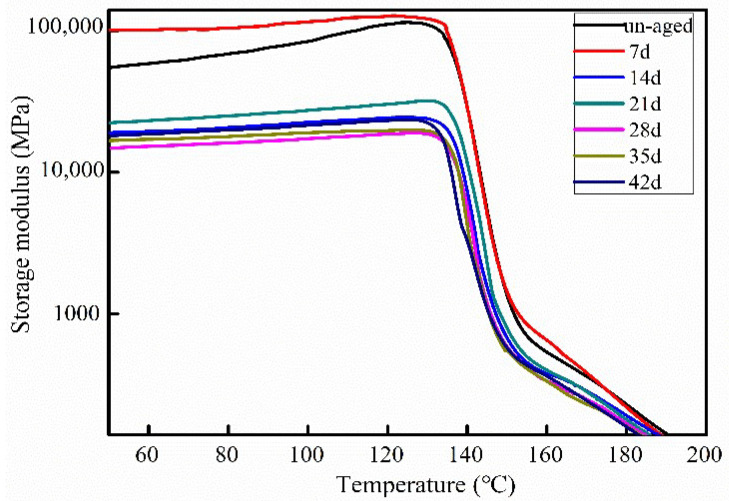
The storage modulus of CF/PC composites at different aging time.

**Figure 8 polymers-12-02453-f008:**
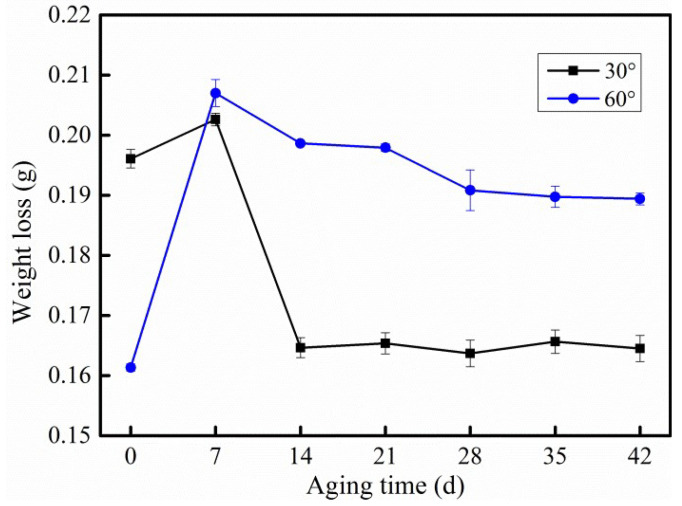
Sand erosion test of CF/PC composites, 1 min, 50 psi.

**Figure 9 polymers-12-02453-f009:**
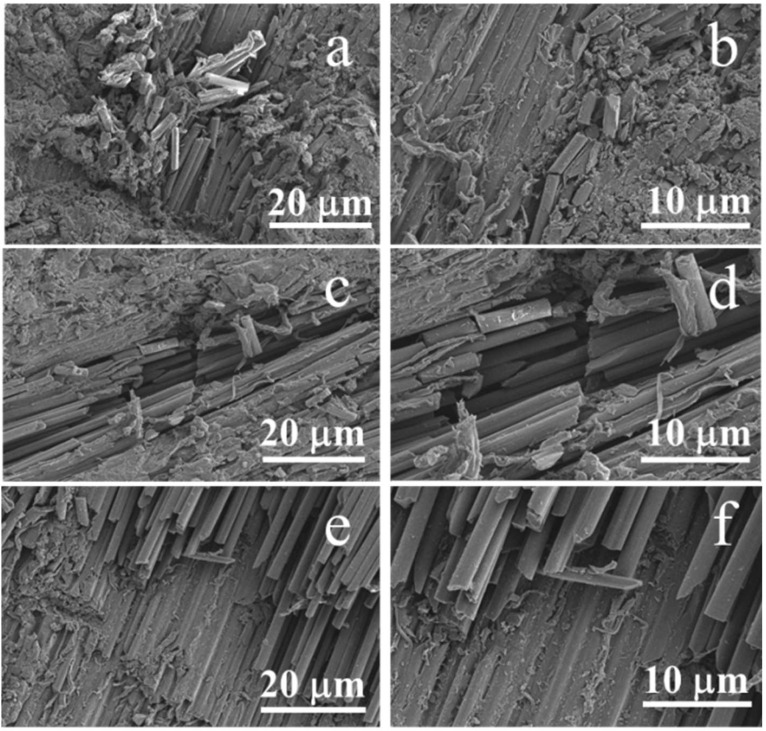
Surface topography of CF/PC composites after erosion. (**a**,**b**) un-aged composites surface after erosion at an angle of 30°,(**c**,**d**) hydrothermal aged for 7 days composites surface after erosion at an angle of 30°, (**e**,**f**) hydrothermal aged for 42 days after erosion at an angle of 30° [29].

**Figure 10 polymers-12-02453-f010:**
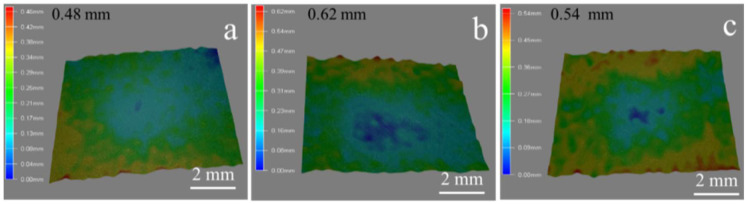
Three-dimensional picture of CF/PC composites at same experimental conditions after sand erosion. (**a**) un-aged composites surface after erosion at an angle of 60°, (**b**) hydrothermal aged for 7 days composites surface after erosion at an angle of 60°, (**c**) hydrothermal aged for 42 days after erosion at an angle of 60°.

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
