# Peer review of "Effects of Hydrothermal Aging of Carbon Fiber Reinforced Polycarbonate Composites on Mechanical Performance and Sand Erosion Resistance"

_polymers, 2020, doi:10.3390/polym12112453_

Round 1

Reviewer 1 Report

The manuscript requires extensive rewriting.  The authors may consider to use the help of a proficient English speaker.   The list of comments includes several examples, but it is not complete. Additionally, there is missing or incomplete information in some sections, and some statements requires further experimental support.

Specific comments are listed below.

Lines 12-13: " ...have attracted numerous attentions ..."  Consider using " ...have attracted attention..."

Line 15: "... the composites were suffered from hydrothermal aging to..."  Consider using: "... the composites were subjected/exposed to hydrothermal aging to..."

Lines 17-18: " It was demonstrated that the peak ... the seventh day." Revise wording.

Lines 56-57: "The underlying molecular mechanisms changes ... property were surveyed." Revise.

Lines 63-64: "The Makrolon series of Polycarbonate (PC) was purchased from ..."  Makrolon is the name of the brand, but you must include the code name of the particular polymer used, for the reader to know specifically which polycarbonate have you utilized in the study.

Lines 62-67:  The description of the procedure is incomplete.  According to Figure 1, eight sheets of the carbon fibers were disposed at different angles, but there is no indication of how the polycarbonate (PC) was incorporated in the composite. Besides, what is the estimated % of carbon fibers in the composite?  Is the preparation the same one used in your previous publication [Mater. Res. Express7(2020)045305,    https://doi.org/10.1088/2053-1591/ab6fa6]. If so, check that description, which was better than the one presented in this manuscript.

Line 69:  "Samples were divided into seven groups and treated in deionized water at 80 oC."  Were the samples immersed in deionized water?

Lines 72-80:  Were the specimens used in the water absorption tests the same that were used in the hydrothermal aging study.

Line 82: "...to study the tensile property..."  Revise ("tensile properties").

Line 84: "The bend property were conducted..." Revise wording.

Line 86: "...for each parts of..."  Revise wording.

Lines 87-90: Revise and rewrite the whole paragraph.

Line 96: "...of the samples with the accelerated by high pressure gas."  Revise wording.

Line 103: "...specimens’ surface aging in different time..."  Revise.

Lines 104-105: "...on surfaces of samples prior to make them more conductive and clear observation."  Revise wording.

Line 116: "Figure 2. Water absorption line of CF/PC composites"  Delete "line".

Lines 117-118: "...the 42 days hydrothermal aging specimen surfaces (Fig. 3 b) appeared deep crack..."  You may want to consider using: "...after 42 days, the surfaces of the hydrothermally aged samples (Fig. 3b) showed deep cracks ..."  You may prefer other alternatives, but in any case, consider rewriting the sentence.

Line 119: "...molecules entered CF/PC composites and possessed a certain volume..."  Revise.

Line 123: "Tensile property analysis"  Revise ("porperties").

Line 124: "The tensile property of CF/PC in different aging time was displayed in Fig.4."  Consider using: "Representative tensile behavior/response of CF/PC at different aging times is displayed in Fig.4."  or "Figure 4 shows the representantive..."

Line 125: "The elongation at break ...decreased as the aging time."  Consider replacing "as" by "with".  In the displayed Figure, your statement is not true for the sample aged for seven days. Is this a representative curve?  Was the average elongation at break increased significatively after 7 days?

Instead of the plot for the maximum load consider including a plot for the Young's modulus of the material and perhaps also the elongation at break.  You could also summarize those results in a Table of mechanical properties.

Lines 126-127: " A higher crosslinking density ..."  The initial polycarbonate is not a crosslinked polymer, thus, what do ou mean by "higher" crosslinking density?  Additionally, what are you suggesting as the mechanism for crosslinking of the PC?

Lines 131-132: "... the tensile property of CF/PC decreased rapidly and reached to the balance as the increasing of aging time."    Revise wording.

Lines 132-133: " the tensile strength ...were ..."  Replace "were" by "was".

Line 137: " Hhydroxyl"  Correct the typographical error.

Line 146: "... flexural strain declines as the aging time..."  Consider using: "... flexural strain declines with the aging time..." or "... flexural strain declines as the aging time increases..."

Lines 146-147: "...indicating a ductile-brittle transition of CF/PC composites."  The strain at break is reduced, but the change is from 2.8% to 2.25% (approximately, as read from the plot) and in both cases the strain is rather low and not indicative of a ductile-fragile transition.  Do you have SEM images of the fractured surfaces in these tests to support that claim?  If the main support for the claim are the erosion results, you should mention that they are presented in another section of the manuscript.

Line 150: "...the matrix of CF/PC would shrink when aged at 80 oC..." What are the results that support this claim? Have you actually observed dimensional changes? or have you found changes in the density of the materials?

Line 160: "This result was consistent with the structure shown in Fig. 6."  Figure 6 shows the results obtained in bending tests; it does not show any structure image or scheme.

Line 182: Figure 9b has already appeared in a previous article of your group:

Mater. Res. Express7(2020)045305,    https://doi.org/10.1088/2053-1591/ab6fa6

Thus, some permission text or citation should be included in the legend.   Additionally, also include the angle of erosion.

Line185:  "...CF/PC composites with un-aged ..."  Revise.

Lines 199-200: " Under the action of hot water aging, polymer chain breaking and untangling." Revise.

Line 202: " Combined with Fig.8 and Fig. 9 (e, f), we found the weight loss of the CF/PC is less."  What do you mean?  "less" than what?

Line 208: "...the surface of non-aging samples ..."  Replace "non-aging samples" by "non-aged samples" or " unaged samples".  Revise the manuscript, because that expression appears more than once.

Line 221: " The moisture rate..."   Replace by " The moisture absorption rate..." or " The rate of moisture absorption..."

Line 222: "And it stays the plateau state..."  Revise wording.

Line 223: "The tensile property demonstrates a ..."  Revise.

Author Response

Please see the attachment, thank you very much.

Reviewer 2 Report

The authors will find some remarks on the proposed paper below. 

From a global point of view, this paper is the sum of several experimental results that are interesting but no analysis of the microsctrutural modification is made to explain the reasons of the observed macroscopic behaviour.

Secondly, what is the objective of this paper? If it is to demonstrate the potentiality of CF/PC to be a relevant solution to a technical problem, the functionnal specifications of this problem must be defined in the introduction part.

line 1: the authors conclude their bibliographic synthesis with the importance of working on the erosion of CRFPs, but there is no bibliographic reference or any information on this phenomenon in their bibliography.

figure 2: sorption graphs are rather drawn according to the square root of time

line 120: do cracks appear after the drying step at 25°C or they are already present during the aging step ?

figure 4: on figure 4-a the maximum stress for 7d aging sample is around 500 MPa while it is more than 570 MPa on figure 4-b

figure 4-c: what is the interest of this graph? Material mechanic is based on stress, not on load that is an structural value.

figure 4-c: how is defined the strain plotted on this graph? Strain at maximum stress? Strain at ultimate stress?

figure 6-b: is this graph of the maximal load or the maximal stress? If it is maximal load, what is the interest?

figure 10: the same color scale must be used for the 3 figures.

line 224: all the tests concluded to a peak at the 7th day of aging except for flexural test with a peak at the 14th day. Is it due to a real difference of behaviour and what is this difference? Or is it due to the frequency of the measurements which does not allow to determine precisely when this peak takes place?

Author Response

(The authors gave the same response as above.)

Reviewer 3 Report

This manuscript needs essential modifications as follows:

  • Please write the manuscript in reporting style or using passive sentences (not active sentences, e.g. we, I, etc), see Lines # 19, 132, and 202. Quality of English expressions is not appropriate and mal-use of vocabulary is found throughout the manuscript, see Lines # 81 (Mechanical instead of Mechanics), and 86 (groups instead of parts). Line # 160, what is this? "This result was consistent with the structure shown in Fig. 6. ". Therefore, the English language must be improved throughout the manuscript.
  • What are the relations between Fig. 4.b & Fig. 4.c, and Fig. 6.a & Fig. 6.b. Therefore, the calculations of tensile strength and flexural stress must be added. Fig. 6.c, which Modulus and how it is calculated?. Moreover, where are the deflection measurements?
  • The pictures of the fracture surfaces of tensile and flexural specimens should be added.
  • The relation between the flexural strength and tensile strength should be discussed.

Author Response

(The authors gave the same response as above.)

Round 2

Reviewer 1 Report

The manuscript has been improved, but it still requires revision.  Although most of the comments are related to the use of the language, they should be addressed because in some cases the analysis of the data is compromised (some paragraphs are not understandable).

Comments are listed below.

Line 22: "... scanning electronic microscope (SEM)..."  Replace "electronic microscope" by "electron microscopy".

Line 31: "Batuwitage et al. [7] researched that the strength of CFRP..." Replace "researched" by "reported".

Line 39: "... molecules caused the matrix to expand..." Replace "caused" by "causes".

Lines 44-45: "... sand erosion is widely concerned due to its serious failure, performance degradation and high maintenance costs."  Revise wording.

Lines 45-46: " helicopter rotor blades and high-speed vehicles, whose surfaces are usually exposed to the air, contain sharp and solid particles."   Are the sharp, solid particles on the surface of the rotor blades or in the air?  Your statement implies that they are in the blades.

Lines 49-51: " Lu et al. [24] researched the impact ...decreased the impact property of CF/PC composites."  Consider using "Lu et al. [24] investigated the impact behavior of unidirectional CF-reinforced epoxy resin compounds after hydrothermal aging and found that the entry of water molecules decreased the impact resistance of CF / PC compounds."

Lines 64-65: " Expectantly, CF/PC material is chosen as the surface material, hydrothermal aging protection 64 must be done, in order not to affect the product performance. " Do you mean: " It is expected that, in the case that CF / PC material is chosen as surface material, hydrothermal aging protection will be applied, so as not to affect the product performance."

Line 74: "... tapes were marshaled for 0°/90°/..."  Do you mean that they were assembled at angles of....?

Lines 79-80: " The aging time for seven parts is 0, 7, 14, 21, 28, 35 and 42 days." Revise wording.

Line 94: " The bend properties were conducted using INSTRON 5585 machine..."  Use " The bending/flexural properties were measured..." or " The bending tests were conducted..."

Line 96: Do you mean that you used seven replicates for each sample?

Lines 98-101: "The DMA analyzer (Q800) was applied ... were conducted."  Consider using "The DMA analyzer (Q800) was used to characterize the dynamic thermo-mechanical properties of un-aged and hydrothermally aged CF/PC specimens. The single cantilever mode was used during the test. The specimens with size of 30 mm × 12 mm × 2 mm were heated from 50oC to 200 oC at a heating rate of 3 oC /min with a frequency of 1 Hz. At least five specimens were scanned for each sample."

Line 113: "specimens aging in different time..." Do you mean "specimens aged during different times"?

Figure 2.  Add two small arrows or dashed lines in the figure indicating "5days" and "7 days", since your plot is given in square root of time in hours, while the text discusses the water absorption in days.

Lines 129-130: "Water molecules entered ...and take up a certain space."  The sentence begins using past tense ("entered"), so the second verb should be "took".  You may consider to change that part to: "...and occupied additional volume".

Lines 137-138: "A tighter construction owing to ..."  Consider replacing "construction" by "structure".

Lines 145-146: " the cohesion of water molecules within composites ..." What do you mean?  Physical bonding to the polymer?  Presence of sorbed water?

Line 158: "the tensile strength and flexural strength value is very close."  Replace "value" by "values" and "is" by "are".

Line 160: Do you mean: "..lower part (subjected to tensile stress) "?

Line 161: "it is shows that..."  Replace by "it is shown that".

Line 170: "...owing to the inflation effects..."  I believe that you meant "swelling", not "inflation".  Revise.

Line 190: "The water molecules immersed composites for long time would make it brittle."  Revise wording.

Lines 202-203: "a few fibers are exposed and break." Replace "break" by "broken".

Line 204: "...deforms scratches and pits on CF/PC surfaces..." Do you mean that"deforms, scratches and forms pits"?

Line 206: "CF/PC composites departure."  What do you mean by "departure"?

Line 215: "the small molecules at the end of the polymer chain degrade..."  What do you mean by "small molecules"?  Are you referring to the terminal segments (chemical groups?) of the PC chain?

Line 219: "...wan the lowest..."   Revise and correct the typographical error ("was").

Lines 237-238: "The moisture absorption rate of CF/PC composites grows linearly with aging time."  Actually,  at short times, moisture absorption grows linearly with the square root of time.

Author Response

Thank you for your careful review.

Reviewer 2 Report

You can find my comments to the first response of the authors in the attached file.

Author Response

Thank you for your detailed illustration.

Reviewer 3 Report

The authors have successfully addressed all my comments.  Therefore, I recommend the publication of this manuscript.

Author Response

Thank you for your encouragement and thank you again for your questions.

Round 3

Reviewer 2 Report

The paper is now admissible for publication in the journal Polymers.